# The Cell-Wall β-d-Glucan in Leaves of Oat (*Avena sativa* L.) Affected by Fungal Pathogen *Blumeria graminis* f. sp. *avenae*

**DOI:** 10.3390/polym14163416

**Published:** 2022-08-21

**Authors:** Veronika Gregusová, Šarlota Kaňuková, Martina Hudcovicová, Katarína Bojnanská, Katarína Ondreičková, Beáta Piršelová, Patrik Mészáros, Libuša Lengyelová, Ľudmila Galuščáková, Veronika Kubová, Ildikó Matušíková, Daniel Mihálik, Ján Kraic, Michaela Havrlentová

**Affiliations:** 1Department of Biotechnology, Faculty of Natural Sciences, University of Ss. Cyril and Methodius, 917 01 Trnava, Slovakia; 2National Agricultural and Food Centre, Research Institute of Plant Production, 921 01 Piešťany, Slovakia; 3Department of Botany and Genetics, Faculty of Natural Sciences and Informatics, Constantine the Philosopher University, 949 01 Nitra, Slovakia; 4Department of Ecochemistry and Radioecology, Faculty of Natural Sciences, University of Ss. Cyril and Methodius, 917 01 Trnava, Slovakia

**Keywords:** *CslF* genes, leaf pathogen, biotic stress, glucanhydrolases, photosynthetic pigments

## Abstract

In addition to the structural and storage functions of the (1,3; 1,4)-β-d-glucans (β-d-glucan), the possible protective role of this polymer under biotic stresses is still debated. The aim of this study was to contribute to this hypothesis by analyzing the β-d-glucans content, expression of related cellulose synthase-like (*Csl*) *Cs1F6*, *CslF9*, *CslF3* genes, content of chlorophylls, and β-1,3-glucanase content in oat (*Avena sativa* L.) leaves infected with the commonly occurring oat fungal pathogen, *Blumeria graminis* f. sp. *avenae* (*B. graminis*). Its presence influenced all measured parameters. The content of β-d-glucans in infected leaves decreased in all used varieties, compared to the non-infected plants, but not significantly. Oats reacted differently, with Aragon and Vaclav responding with overexpression, and Bay Yan 2, Ivory, and Racoon responding with the underexpression of these genes. Pathogens changed the relative ratios regarding the expression of *CslF6*, *CslF9*, and *CslF3* genes from neutral to negative correlations. However, changes in the expression of these genes did not statistically significantly affect the content of β-d-glucans. A very slight indication of positive correlation, but statistically insignificant, was observed between the contents of β-d-glucans and chlorophylls. Some isoforms of β-1,3-glucanases accumulated to a several-times higher level in the infected leaves of all varieties. New isoforms of β-1,3-glucanases were also detected in infected leaves after fungal infection.

## 1. Introduction

Cellulose is the main polysaccharide of plant cell walls and cellulose microfibrils form the stiffening rods on the wall [1]. The non-cellulosic wall polysaccharides of dicotyledonous plants are rich in pectic polysaccharides and xyloglucans, whereas the walls of monocotyledonous plants of the family *Poaceae*, including cereals and grasses, have relatively small amounts of pectic polysaccharides and xyloglucans, but proportionately higher amounts of heteroxylans [2]. In addition, the cell walls of some grasses contain significant amounts of (1-3; 1-4)-β-d-glucans (β-d-glucans), natural polymers that are not widespread in other plant species [3]. β-d-glucans are linear polymers of glucose units linked by about 75% β-(1-4) and 25% β-(1-3) bonds and arranged in an irregular but non-random sequence such that no consecutive β-(1-3) bonds occur [4,5]. Interest in cereal β-d-glucans has increased after their acceptance as bioactive and functional ingredients in a healthy diet [6,7,8] for human and animals. Nevertheless, β-d-glucans play an important role in the structure and functionality of cereal cell walls [5,9]. Several studies have addressed the protective role of this biopolymer during fungal infection. Increased pre-infection β-d-glucans content was shown to reduce the overall pathogen activity, expressed by the reduced presence of fungal DNA as well as reduced deoxynivalenol (DON) content after the artificial inoculation of oat with *Fusarium* (*F*.) *graminearum* and *F. culmorum* [10] or with *F. graminearum* in barley [11]. The antioxidant activity of β-d-glucans was also described [12], whereby antioxidant compounds are an accepted factor of the plant’s defence system to cope with biotic aggressors such as fungal pathogens [13,14]. The increase or decrease in the biosynthesis and associated content of β-d-glucans is conditioned by the expression of plant genes involved in both β-d-glucan synthesis and its degradation. After the exposure of plants to external factors such as pests or pathogens, the β-d-glucan metabolism as well as the entire β-d-glucan turnover are affected and become much more complicated. β-d-glucan synthesis is mostly studied in barley and oat, and its mechanism and genes are basically the same. The most important gene involved in β-d-glucan biosynthesis in oats is cellulose synthase-like (*Csl*) gene *CslF6* [15]. Genes from the *CslH* [16] and *CslJ* [17] subfamilies, specific to the *Poaceae*, were later found to relate to the β-d-glucan synthesis. However, the functions of many members of each of these subfamilies are yet to be characterised [18]. *CslF6*, *CslH*, and *CslJ* have been proved to be directly involved in β-d-glucan synthesis by either mutant knockout/down of *CslF6* transgenic overexpression in barley grain [19,20] via an endosperm-specific promoter [1] or heterologous expression in either *Arabidopsis thaliana* or tobacco leaves (*Nicotiana tabacum* L.) [18]. The *CslF6* gene is transcribed at relatively high levels among all predicted β-d-glucan synthase genes in developing and vegetative tissues, but *CslF9* is expressed during early grain development and in root tips [21]. *CslF3* is also highly expressed in root tips and coleoptiles [21,22]. Knowledge of the presence and functions of enzymes involved in the dynamics of β-d-glucan turnover is still fragmentary and focused on selected species. However, genes *GlbI* and *GlbII* encoding endohydrolases that cleave β-d-glucans have been described, mostly in the context of seed development [21,23].

When a plant is infested with a fungal pathogen, the composition of its cell wall is also important. The inhibition of cell wall synthesis and the lysis of existing cell walls of fungi represent key strategies used by plants to fight against fungal invasion [24]. Enzymes of β-1,3-glucanases (glucan endo-1,3-β-glucosidase, EC 3.2.1.39) are an important group of hydrolytic enzymes belonging to the pathogenesis-related (PR) proteins family 2 [25]. β-1,3-glucanases predominantly recognise and cleave β-1,3-linkages, and β-1,3;1,4-glucanases recognise and cleave β-1,3;1,4-linkages (reported only in monocots of *Poaceae*). A variety of β-1,3-glucanases with different primary structures and sizes, cellular localisation, and patterns of regulation have been described in a range of plant species as a response to pathogen attack [26]. Molecular studies described the early up-regulation of these enzymes in resistant cultivars of wheat (*Triticum aestivum* L.) [27], but not in susceptible ones during the first days after infection [28,29,30]. Enhanced activity of β-1,3-glucanases has been reported in barley (*Hordeum vulgare* L.) after infection with *B. graminis* (DC) Speer f. sp. *hordei* [31]. They probably target *B. graminis* conidial cell wall polysaccharides, which consist of 63.1% glucosyl residues dominantly linked by 1,3-linkages [32]. *B. graminis* is an obligate pathogen and causal agent of powdery mildew on cereals, one of the most common and devastating diseases of plant species from more than 100 genera of the family *Poaceae* [33,34,35], especially in the northern hemisphere [34,36]. Regarding oat (*Avena sativa* L.), powdery mildew is one of the most harmful diseases causing yield losses up to 32% [37,38]. Recently, studies from China and the mountain region of northern India [39,40] reported its occurrence and negative impact on oat cultivation. Usually, *B. graminis* colonies are developed on the leaf surface, and the mycelium penetrates the host cell wall.

The storage function of β-d-glucans in the plant seeds as a source of energy (glucose) for developing sprouts has been suggested [1], but the protective role of individual cell wall polysaccharides against biotic or abiotic stresses is mostly unexplored. Regarding the possible protective role of β-d-glucans, the aim of this study was to evaluate: (i) the expression of *Cs1F6*, *CslF9*, and *CslF3* genes related with synthesis of β-d-glucans, (ii) the relative activity and structure of β-1,3-glucanases with the role in plant defence against fungal pathogens, but also in the degradation of the β-d-glucans and (iii) content of photosynthetic active pigments in oat leaves infected by the fungal pathogen *B. graminis*.

## 2. Materials and Methods

### 2.1. Plant Material and Pathogen Infection

Five oat (*Avena sativa* L.) varieties (Aragon, Bay Yan 2, Ivory, Vaclav, Racoon) were selected based on the leaf β-d-glucans content from previous pre-screening (data not shown). None of these varieties contain any of the resistance genes, *Pm1*, *Pm2*, *Pm3*, and *Pm6*, against *B. graminis* determined using the leaf segments inoculated by pathogen isolates [41]. Seeds were planted in pots, at 10 plants per pot; one set was used as the control and the other set was used for artificial infection with *B. graminis*. Cultivation conditions were as follows: photoperiod—day/night 16 h/8 h, temperature 15–24 °C, air humidity 29–64% for infected samples and 67–82% for control samples. Plants were cultivated to the third leaf stage and in this stage, they were inoculated with spores (1000–2500 spores per cm^2^) of *B. graminis* by using settling tower. Spores were dispersed in turbulent air in a settling tower according to the method presented in [42]. The density of inoculation was determined on Petri dishes with agar medium which were placed on the top of inoculated pots. The cultivation of infected plants was carried out separately from the control plants for four weeks. Leaves from 4-week-old plants were obtained from various plant samples from the same pot and dried at 40 °C for the analysis of β-d-glucans. Moisture level was determined after three days using the laboratory moisture analyser Sartorius MA 45 (Sartorius, Göttingen, Germany). Leaf samples for RNA isolation were isolated from young leaves and stored in 95% (*v*/*v*) ethanol in RNA-free tubes at −85 °C. Fungal spores were also separately cultured on oat leaves and after 20 days were collected and used for the analysis of acid and neutral glucanases on PAGE (2.4).

### 2.2. Determination of β-d-Glucans in Leaves

The total content of β-d-glucans was analysed using the mixed-linkage β-d-glucans assay kit (Megazyme, Bray, Ireland). Control and infected samples of dry leaves were milled and passed through a 0.5 mm sieve using an ultra-centrifugal mill (ZM 100, Retch, Haan, Germany). A 100 mg aliquot of each sample was wetted with 0.2 mL of 50% ethanol in a tube. Sodium phosphate buffer (4 mL, 20 mM, pH 6.5,) was added and the sample was vortexed and incubated in a water bath (100 °C for 3 min, then 50 °C for 5 min). Lichenase (0.2 mL, 10 U) was added to the tube and then the reaction mixture was incubated for 1 h at 50 °C with regular vigorous stirring (4 times every 15 min) on a vortex mixer. Sodium acetate buffer (5.0 mL, 200 mM, pH 4.0) was added and the content was vigorously mixed in a vortex mixer. After 5 min of equilibration at room temperature and centrifuging (1000× *g*, 10 min, 24 °C) 0.1 mL aliquots were dispensed into three 12 mL test tubes. β-glucosidase (0.1 mL, 0.2 U) in 50 mM sodium acetate buffer, pH 4.0, was added to two of these tubes (reaction), whereas the enzyme was replaced by 0.1 mL of 50 mM sodium acetate buffer, pH 4.0, in the third tube. All tubes were incubated at 50 °C for 10 min. The GOPOD Reagent (3.0 mL) was added to each tube and incubated at 50 °C for a further 20 min. The tubes were removed from the water bath and the mixture absorbances were measured within 1 h at 510 nm against the reaction blank. The absorbance at 510 nm against reagent blank within 1 h was measured. The absorbance readings were converted into β-d-glucan content by using the following formula: β-d-glucan (%) = ΔA⋅FW⋅8.46. F is determined by the ratio of glucose mass (100 μg) and glucose absorbance (100 μg), W is the weight of the sample expressed in mg, ∆A is the extinction after β-glucosidase treatment (reaction) minus reaction blank absorbance, and 8.46 is the conversion factor. Each sample’s β-d-glucan content was calculated as the mean of three replicates and was expressed as a percentage of the dry matter. The content of β-d-glucans was the mean from three replications.

### 2.3. Content of Chlorophylls

The content of assimilation pigments was determined in acetone extracts according to [43]. The measured parameters were as follows: chlorophyll a (Chl_a_), chlorophyll b (Chl_b_), total chlorophylls (Chl_tot_), total carotenoids (Car), and chlorophyll ratio (Chl_a/b_). Leaf tissue (80 mg of fresh weight) was ground in a mortar in 5 mL of cold 80% (*v*/*v*) acetone, filtered through a sintered-glass funnel (Sinter No. 3), and diluted to 10 mL with 80% (*v*/*v*) acetone. The concentration of photosynthetic pigments was determined from absorbance peaks at 470, 663, and 646 nm, respectively, using the spectrophotometer UV-2601 (Shimadzu, Kyoto, Japan). Acetone was used as a blank. Pigment concentrations were determined by the following formulae:Chl_a_ = 12.21 A_663_ − 2.04 A_646_(1)
Chl_b_ = 20.13 A_646_ − 4.19 A_663_(2)
Chl_tot_ = 7.05 A_663_ + 18.71 A_646_(3)
Chl_a/b_= Chl_a_/Chl_b_(4)
Car = (1000 A_470_ − 3.27c_a_ − 104c_b_)/229(5)

The content of pigments was determined in 3 or 4 replicates in each variant of the experiment.

### 2.4. βGlucanase Analysis

Crude protein extract was isolated from 3–4 plants’ leaves using an extraction buffer containing 0.1 M (*w*/*v*) sodium acetate (pH 5.0) and 1 mM phenylmethylsulfonyl fluoride. Samples (0.5 g) were ground in a mortar with liquid nitrogen, transferred into extraction buffer, and homogenised by vortexing. Insoluble debris was removed by centrifugation at 14,000 rpm, 4 °C, for 15 min. Removed supernatant was centrifuged again at 14,000 rpm, 4 °C, for 10 min. Protein concentration was determined by Bradford assay [44].

Sodium dodecyl sulphate polyacrylamide gel electrophoresis (SDS-PAGE) was performed for separation and in-gel detection of total β-1,3-glucanases, based on the modified procedure described by Pan et al. [45], which was successfully used in several previous works [31,46,47,48,49,50]. Separation was performed in 1.5 mm thick minigels (Mini-Protean Tetra Cell apparatus, BioRad Laboratories, Hercules, CA, USA) according to Laemmli methode [51]. The 12.5% (*w*/*v*) polyacrylamide gels contained 0.25% (*w*/*v*) laminarin (Sigma-Aldrich, Darmstadt, Germany). Laminarin is the preferred substrate for β-d-glucanhydrolases in barley [52]. Protein aliquots (20 µg) were loaded on the gels without any heat treatment in loading buffer containing 0.005% (*w*/*v*) bromphenol blue, 5% (*v*/*v*) β-mercaptoethanol, 4.5% (*w*/*v*) SDS, 20% (*v*/*v*) glycerol, 0.6% (*w*/*v*) Tris, and 0.02% (*w*/*v*) TEMED (N,N,N′,N′- tetramethylethylenediamine, Sigma-Aldrich, Darmstadt, Germany). Electrophoretic separation was run under constant 24 mA, 4 h, at 6 °C for 4 h. After electrophoresis, proteins were re-natured by shaking the gels in 50 mM (*w*/*v*) sodium acetate buffer (pH 5.2), 1% (*v*/*v*) Triton X-100 for 1 h. Molecular weights of proteins were estimated by co-electrophoresis of protein ladder (SpectraTM Multicolor Broad Range Protein Ladder, Thermo Scientific, Shanghai, China) ranging from 10 to 260 kDa.

Acidic/neutral and basic/neutral glucanases in extracts were separated in 11% (*w*/*v*) native (without SDS) polyacrylamide gels containing 0.25% (*w*/*v*) laminarin as a substrate [53]. The loading buffer contained 50% (*v*/*v*) glycerol and a dye, namely 0.005% (*w*/*v*) bromphenol blue in the case of acidic and neutral proteins and 0.01% (*w*/*v*) methylene blue in the case of basic and neutral proteins. Gels were run at 24 mA, 5 h, at 6 °C.

Enzymatic activities of glucanhydrolase isoforms were detected as described previously [45]. Electrophoretic gels were incubated in 0.5 M (*w*/*v*) sodium acetate buffer (pH 5.2), 2 h, at 37 °C, and ubsequently were fixed in 7% (*v*/*v*) acetic acid in 20% (*v*/*v*) methanol for 5 min and washed with distilled water. The staining was performed by boiling the gels in 0.1% (*w*/*v*) 2,3,5-triphenyltetrazolium chloride (Sigma-Aldrich, Saint Louis, MO, USA) in 1 M (*w*/*v*) NaOH in a water bath. After 5–10 min, red bands indicating β-1,3-glucanase activity appeared. The gels were placed in 7% (*v*/*v*) acetic acid, scanned (Epson Perfection V600, Epson, Suwa, Japan), and images were processed using Scion Image software (https://scion-image.software.informer.com accessed on 3 February 2021). The relative activity of β-1,3-glucanase isoforms was determined based on the background-corrected mean density of fractions (evaluated in pixels) and was expressed as a multiple of the corresponding control sample.

### 2.5. Expression of CslF Genes

The expression level of the related genes *CslF3*, *CslF6*, and *CslF9* from the control and infected samples was analysed by means of reverse transcription polymerase chain reaction (RT-PCR) and quantitative PCR (qPCR). Total leaf RNA was extracted from plant samples using 1 g of plant tissues by the TRIzol reagent method (Invitrogen Corp., Carlsbad, CA, USA) [54]. The concentration of isolated RNA was determined spectrophotometrically (Nanodrop 1000 Spectrophotometer, Thermo Fisher Scientific, Waltham, MA, USA). RNA samples were reversely transcribed using The RevertAid First Strand cDNA Synthesis Kit (Fermentas, St. Leon-Rot, Germany). Twenty-five nanograms of RNA was used for the synthesis of cDNA with RevertAid Reverse Transcriptase (200 U/µL) (TermoFisher, Waltham, MA, USA) following the protocol [55]. The cDNA solution was used as a template for qPCR amplification using SYBR Green Master Mix (Applied Biosystems, Foster City, CA, USA) with the specific primers designed for *A. sativa* genes *CslF3* (MG543996.1), *CslF6* (MG543998.1), and *CslF9* (MG544000.1) from known sequences (https://www.ncbi.nlm.nih.gov/ accessed on 10 February 2022) in PrimerExpress (Applied BioSystems, Waltham, MA, USA) (Table 1). The gene for *A. sativa*, elongation factor 1 alpha (*EF-1A*, MH260253.1), was used as an internal control (Table 1) [56,57]. The amplification of all genes was carried out in 25 µL of a reaction mixture containing 12.5 µL of a SYBR^®^ Green PCR Master Mix, 0.150 µM of both primers, 2.5 µL of cDNA, and water to a final volume. qPCR war performed in ABI PRISM^®^ 7000 (Applied BioSystems, Waltham, MA, USA), and the PCR conditions were as follows: 50 °C for 2 min, 95 °C for 10 min, 40 cycles at 95 °C for 15 s and 59 °C for 1 min. The quantification of the gene expression level was performed with the comparative CT method [58].

### 2.6. Statistical Analysis

The standardised skewness and standardised kurtosis were used to determine whether the sample comes from a normal distribution. Values of these statistics outside the range −2 to +2 indicate significant departures from normality. In the case of the data obtained from the experiments, they showed standardised skewness and standardised kurtosis values which were not outside the expected range. Statistically significant differences between control and infected samples were tested using the *t*-test at the 95% confidence interval for the means, and Pearson product–moment correlations between each pair of variables, measuring the strength of their linear relationship, were calculated using the software Statgraphics XVII—X64 (Statpoint Technologies, Inc., Warrenton, VA, USA). Data for variables (average values for Chl_a_, Chl_b_, Car, Chl_a/b_, total chlorophyll, total glucanases, acid and neutral glucanases, and all values for the genes *CslF3*, *CslF6*, and *CslF9*, even partial, from the calculation for the delta-delta Ct method as ∆Ct, ∆∆Ct and 2^-∆∆Ct) were also used for principal component analysis (PCA) using the scores of the first two principal components and scores from all principal components with a measured Euclidean distance with 9999 permutations were used for PERMANOVA analysis using the PAST (PAleontological STatistics) software version 3.19 [59].

## 3. Results

### 3.1. Determination of β-d-Glucans

Generally, oat leaves from control plants and plants infected with *B. graminis* contained trace amounts of β-d-glucans. However, after the infection, the content of β-d-glucans statistically significantly decreased in all oat varieties, and the P-values for Aragon, Bay Yan 2, Ivory, Vaclav, and Racoon were as follows: 0.00134, 0.00044, 0.00024, 0.00073, 0.00179, respectively (Figure 1). The highest decrease compared to the control was in infected Ivory and Vaclav oats (−87.50% and −80.61%, respectively). The other three infected oats (Aragon, Racoon, and Bay Yan 2) had a mutually comparable decrease in the content of analysed β-d-glucans (−52.14%, −49.13%, and −46.72%, respectively).

### 3.2. Chlorophyll Content

Leaves infected by fungus had a generally decreased content of photosynthetic pigments in all oat varieties. The content of Chl_a_ decreased after the infection in almost all samples, whereby in Bay Yan 2, a mild increase was detected (Figure 2). The most sensitive variety was Ivory, where the decrease in Chl_a_ was 55.26% compared to the control. Racoon, Aragon, and Vaclav showed a decrease in Chl_a_ content in the range of 34.38–39.32%. A decreasing trend after the infection was observed in the content of Chl_b_. A decrease of 64.84% was observed in Ivory, while in Aragon, Vaclav, and Racoon, the decrease was from 37.78% to 54.69%. Statistically significant changes in Chl_tot_ were recorded in all oat varieties, except for Bay Yan 2. The ratio of Chl_a_ to Chl_b_ (Chl_a/b_) did not change significantly in all samples, except for Aragon. A statistically significant increase (52.20%) was detected after the infection with *B. graminis* compared to the control. The content of Car decreased significantly after the infection in all monitored oat varieties. The lowest decrease (29.55%) was in Bay Yan 2, and the largest (65.00%) was in Ivory. The content of Car in Vaclav, Racoon, and Aragon was in the range of 39.22–56.25%. Remarkably, in Bay Yan 2, the variety with the highest content of β-d-glucans in the control condition (Figure 2), the lowest changes between control and infected samples in the content of photosynthetic pigments were detected.

### 3.3. Glucanhydrolases

Isoforms of six different sizes (~140, 80, 60, 48, 40, 35 kDa) were detected in oat leaves. The enzymes of 48, 40, and 35 kDa appear to be present in leaves of all tested oat varieties, while the accumulation of some enzymes was influenced by fungal infection (Figure 3). The expression of the 48 kDa isoform was significantly suppressed in all infected samples (Table 2). The relative accumulation of the 40 kDa isoform was decreased by 37% in Racoon but increased by 208% in Bay Yan 2. The relative accumulation of this isoform was not significantly affected in other studied oat varieties. The relative accumulation of the 35 kDa isoform was enhanced in all samples, but significantly only in the leaves of Racoon (Figure 3, Table 2). At least three other isoforms (~140, 80, 60 kDa) were synthetized de novo in response to *B. graminis* infection. The isoforms of 80 and 60 kDa were induced by the pathogen in all of the studied oat varieties, while the 140 kDa isoform appeared specific for infected leaves of Aragon (Figure 3).

When enzymes of different charges were studied separately, up to six isoforms (A1-A6) of acidic and neutral β-1,3-glucanases were detected. Isoforms A1 and A2 were detected only in the infected leaves of some varieties (A1 in Aragon and Racoon, A2 in Bay Yan 2 and Racoon) (Figure 4). The accumulation of A3, A4, and A6 was almost 2 to 18 times higher in infected leaves of all oat varieties compared to control plants. Additionally, eight times higher accumulation was observed in isoform A5 in infected leaves of Bay Yan 2. This isoform was also detected in infected leaves of Ivory, Vaclav, and Racoon (Figure 4, Table 3). A single basic/neutral isoform was reproducibly present in experimental plants; its responsiveness to powdery mildew, however, was not confirmed statistically (Figure 5).

### 3.4. CslF6 Gene Expression

The *CslF6* gene is a major gene involved in the biosynthesis of β-d-glucans in cereals. The Ct-values from RT-qPCR plotted against control (infection-free) growth and fold gene expression were analysed by means of 2^-(∆∆Ct). The highest overexpression of this gene was recorded in Aragon (288% ± 94%) gene expression relative to the control plants, but it was not significant, despite such a difference (*t*-test, *p* > 0.05). Statistically significant differences were not confirmed even in additional statistical analyses, such as the F-test to compare standard deviations and the Mann–Whitney (Wilcoxon) W-test to compare medians (data not shown). On the other hand, the overexpression of *CslF6* observed in Vaclav was statistically significant. Other infected oats showed statistically significant underexpression of this gene related to the control plants (Figure 6).

### 3.5. CslF9 Gene Expression

The gene *CslF9* has a similar function to *CslF6.* It is also involved in β-d-glucans biosynthesis, and quantitative RT-PCR results were analysed in the same way as in the *CslF6* gene. The resulting fold gene expression for *CslF9* had a similar trend to *CslF6*. However, the overexpression of *CslF9* in Aragon and Vaclav was almost two times higher than that of the *CslF6*. Even in this case, Aragon showed the highest expression (545% ± 184%), which was not statistically significant (*t*-test, *p* > 0.05). Significance was detected in additional statistical analyses in the comparison of standard deviations (F-test, *p* = 0.02381), but not in the comparison of medians (W-test, *p* > 0.05). For the *CslF9* gene, Racoon did not show statistically significant underexpression compared to the control plants, and it was 13% ± 7% (*t*-test, *p* > 0.05). The lowest expression rates compared to control plants were observed in Bay Yan 2 and Ivory, at 5% ± 2% and 8% ± 6%, respectively, (Figure 7).

### 3.6. CslF3 Gene Expression

The *CslF3* gene does not have a proven capability for β-d-glucans synthesis, but it may serve as an additional compartment in β-d-glucans biosynthesis. As observed in the *CslF6* and *CslF9* genes, *CslF3* also showed the same trend of reduced expression in Ivory, Bay Yan 2, and Racoon in infected samples compared to the control plants (15% ± 6%, 15% ± 11%, 32% ± 10%, respectively). Standard overexpression was recorded in Aragon and Vaclav (147% ± 2% and 126% ± 17%, respectively). Only one variety (Ivory) showed a statistically significant difference between the control and infected samples in *CslF3* expression (Figure 8).

### 3.7. Relationships among All Evaluated Parameters

The relationships between all monitored parameters were evaluated using Pearson correlation as well as using principal component analysis (PCA). Pearson product–moment correlations between each pair of evaluated parameters, regardless of the specific oat variety, were manifested differently in control conditions and differently in infected oat samples (Figure 9). According to this, there is no relationship between β-d-glucans content and *CslF* genes expression and between β-d-glucans content and pigment content in control varieties. However, this neutral relationship was changed during the infection. The expression of all three *CslF* genes during infection with *B. graminis* was statistically positively correlated with each other, and there was an indication of negative correlation between the β-d-glucans content and *CslF* gene expression, although this trend was not significant. Thus, the higher the gene expression was, the lower β-d-glucans content in the leaves, and there was presumably increased degradation of β-d-glucans. Additionally, the neutral relationship between β-d-glucans content and pigments from the control condition was changed during the infection, becoming a positive correlation; and the higher the β-d-glucans content, the greater the content of Chl_a_ and Chl_tot_ in leaves. A statistically strong positive correlation between the content of individual pigments was observed both in the control conditions and in the infection, and these relationships were unchanged during the infection. Moreover, statistically strong positive correlation was also observed between the *CslF9* and Chl_a/b_. Since this only applies to this one gene, it can be caused by chance or as an artefact of multiple comparisons, and it needs to be verified by further experiments.

The principal component 1 (PC1) separated control oat samples from infected samples (Figure 10). The control samples showed similarity to each other and were divided mainly by the content of pigments (Chl_a_, Chl_b_, Chl_tot_, and Car) and by content of β-d-glucans. PC2 divided two control samples (Racoon and Aragon) from the other controls, and they were located in positive PC2 values along with the parameters for fold gene expression *CslF6* and *CslF3*, which were decisive for the placement of these two varieties in the PCA plot. RT-qPCR data and the presence of various isoforms of glucanases were relevant for the distribution of infected samples, which showed higher variability among themselves. At the same time, the infected Aragon and Vaclav were separated based on PC2 from the other infected varieties in the PCA plot, and this was consistent with the RT-qPCR values, where these varieties showed overexpression of all three genes compared to the control (Figure 6, Figure 7 and Figure 8). Statistical comparison of data from all nine principal components, not only PC1 and PC2 (shown in Figure 10), showed a statistically significant difference between control and infected oat varieties (PERMANOVA, *p* = 0.0082).

## 4. Discussion

The biotic stress caused by *B. graminis* infection manifests itself in the plant in several ways. The aim of this study was to focus on the effect of the artificial fungal infection on the content of cell wall polysaccharide β-d-glucans, and the expression level of genes involved in the biosynthesis of β-d-glucans in oats, including the alteration of photosynthetic pigments content and the presence of glucanhydrolases as a part of the plant defence system. Environmental factors, such as abiotic and biotic stresses, have a meaningful effect on β-d-glucans in cereals. β-d-glucans are one of the storage polysaccharides found in mature grains, but smaller amounts can also be found in stems and leaves [8,60]. β-d-glucans are primarily implicated in the regulation of cell wall expansion due to their high and transient accumulation in young and expanding tissues in the coleoptile and seedling [2,61,62]. In leaves, β-d-glucans provide structural and mechanical support [16,63]. The plant cell wall is one of the factors responsible for plant response to abiotic and biotic stress factors [17] through cell surface hardening [64]. Quick hardening of the cell wall can reduce the success rate of pathogen penetration, and the thickening of the cell wall can be accomplished by β-d-glucans [65,66].

The content of β-d-glucans decreased statistically significantly in our experiment in all oat varieties after the artificial infection with *B. graminis*. Similar findings were observed also in barley and oat seeds [20,21] after artificial inoculation with *Fusarium* sp. The decrease in β-d-glucans may be explained by the presence of β-d-glucanases in fungal pathogens, which enzymatically cleave cereal β-d-glucans and utilise them as a source of glucose for its growth [67]. β-d-glucans are present in the cell wall of monocotyledonous plants, but are also abundant in bacterial and fungal species. They characterise potential damage-associated molecular patterns (DAMPs) and microbe-associated molecular patterns (MAMPs) in monocots and dicots, respectively [68]. During plant–pathogen interaction, enzymes of the pathogen are produced to cleave cell walls into the extracellular space to release small oligosaccharides [69]. Thus, the carbohydrate-rich cell walls of plants and their pathogens represent a source of post-target DAMPs and MAMPs. Recently, several cell wall-derived DAMPs and MAMPs have been identified [70,71,72], but it is possible that a considerably larger number of cell wall-derived ligands from plants can be found. During infection, plant β-d-glucans can be cleaved by glucanases to oligosaccharides, and can activate “immune” responses. Thus, β-d-glucans’ oligosaccharides can be classified as DAMPs in monocotyledonous species [72,73,74]. Cell wall oligosaccharides in rice were degraded by glycosyl hydrolase family GH12 endoglucanase MoCel12A/B during *Magnaporthe oryzae* fungal infection. The result of degradation was specific oligosaccharides being produced, such as trisaccharide β-d-cellobiosyl-glucose and tetrasaccharide β-d-cellotriosyl-glucose, which can be one of the factors activating immune responses in rice plants. Notably, these oligosaccharides contain a β-1,3-1,4-glucan backbone derived from hemicellulose and primarily present in *Poaceae* species [75].

The expression of genes involved in β-d-glucan biosynthesis was monitored in our study. Considerable overexpression of *CslF6*, *CslF9*, and *CslF3* genes was recorded in two oat varieties (Vaclav and Aragon) after the infection. However, the other three varieties (Bay Yan 2, Ivory, and Racoon) showed reduced expression of these genes in infected conditions compared to the control. *CslF6* is included in a group of genes that are expressed in plants during stress, and its expression during infection is decreased [76]. There is information available concerning the response of *CslF6* to biotic stress such as pathogens in terms of carbon metabolism. It appears that with the loss of the function of *CslF6*, sufficient biosynthesis of the cell wall polysaccharide is not possible. It has been documented that those plants can sense and monitor disturbances or changes in the cell wall integrity and activate signalling pathways in response to these wall feedback stimuli [77,78,79]. The absence of biosynthesis by losing the function of *CslF6* and absence of β-d-glucans in the cell wall triggers a signalling cascade that activates stress- and defence-related gene expression and eventually a cell death program [20].

Although *CslF6* transcripts are the most abundant of all predicted β-d-glucans synthase genes in developing grains and many vegetative tissues [21], the transcription of other *CslF* family genes can also be observed during different developmental stages in the plant [21,22]. *CslF9* is mainly expressed during early grain development and in root tips [21], while *CslF3* is highly expressed in root tips and coleoptiles [22]. From our results, the expression of *CslF3* and *CslF9* can also be observed in oat leaves. An increased expression of *CslF3* and *CslF9* genes was observed in nematode-infested barley, whereas *CslF6* transcripts were the most abundant before nematode inoculation and decreased immediately thereafter, exhibiting relatively stable levels across the time course, independent of cultivar or inoculation [60].

Based on the results of our study, it should be emphasised that all changes in the expression of the genes *CslF6*, *CslF9*, and *CslF3* did not statistically significantly affect the content of β-d-glucan in leaves of oats infected with the fungal leaf pathogen *B. graminis*. However, the presence of pathogens in leaves changed the relative ratios in the expression of *CslF6*, *CslF9*, and *CslF3* genes, compared to control, non-infected plants. Their neutral relationships became a statistically significant negative correlation. This could indicate that the pathogenic infection interferes with the complex mechanism of β-d-glucan synthesis

Artificial infection with *B. graminis* caused in our experiment changes in photosynthetic pigments, whereas in Car, the most significant decrease in all varieties after the infection was observed. Ivory was the most sensitive of all monitored oat varieties regarding the change in photosynthetic pigments (Chl_a_, Chl_b_, and Car). On the other hand, Bay Yan 2, with the highest content of β-d-glucans in the control conditions, showed the lowest changes in the content of Chl_a_, Chl_b_, and Chl_tot_ and a significant decrease in Car after the infection compared to other studied varieties. Decreased Chl_a_ and Chl_b_ levels after fungal infection were also reported by others [80]. The main growth of the fungal pathogen with accumulated white mycelium takes place on the leaf surface [81] and penetrates the leaf tissue, leading to more severe damage of the photosynthetic apparatus and chlorophyll breakdown. Some studies [82,83,84,85] focused on linking changes in the reflex spectrum to plant responses to physiological stress, where they also observed a decrease in chlorophylls during powdery mildew infection. It is well known that most pathogen invasions lead to a decrease in host photosynthetic rate [86,87]. However, various authors have closely followed the changes in photosynthesis during pathogen infection in the host and found a decrease in photosynthetic capacity in the early stages of infection [88,89,90,91,92]. Noteworthily, shortly after *B. graminis* infection, resistant barleys show a rapid decrease in photosynthetic capacity in comparison to susceptible plants, probably due to an alteration in source–sink relations and carbon utilisation in resistance pathways [88].

Plants actively remodel the structure and content of β-d-glucans in tissues [22]. In vegetative tissues, functions of β-d-glucanhydrolases have been extended to defence after pathogen attack and to cooperate in plant resistance. Especially different β-1,3-d-glucanhydrolases can act as a pre-formed non-specific defence against a wide range of pathogens by degrading their cell wall in plant´s extracellular space [93]. In addition, these enzymes can release oligosaccharides from the surface of the fungus, which initiate further downstream defence responses [94,95]. Degradation of β-d-glucans have been shown to activate genes including those involved in the synthesis of β-1,3-glucanases [96]. Most class I β-1,3-glucanases are basic proteins localised in the cell vacuole, while most class II, III, and IV β-1,3-glucanases are acidic proteins which are secreted into the extracellular space. Despite extensive research, reports on their involvement after attack by *B. graminis* and in oat are scarce. Of the set of isoforms detected in oat leaves of all oat varieties, three isoforms are permanently present, with likely different biological functions. The activity of the 35 kDa isoform was enhanced after the infection with *B. graminis* in all oat varieties. Similar induction after fungal attack was observed in barley [97] and other plant species [98,99], probably as an early general response of cereals to infection by fungal pathogens [97]. In contrast, reduced activity of the 48 kDa isoform can either coincide with strengthening the plant cell wall or with the modulation of stomatal passability for fungus [100,101,102]. Elevated early callose deposition has been linked with penetration resistance to powdery mildew in barley [103] as well as *Arabidopsis* [100]. The inhibition of glucanhydrolases might also contribute to the prevention of assimilate escape [101]. The possible role of the 40 kDa enzyme with variable response to pathogens in different varieties is unclear. The larger isoforms of detected β-d-glucanhydrolases (~60, 80, and 140 kDa) appeared only in infected leaves. We propose them as the best candidates for some role in oat defence against pathogen. Higher molecular weight isoforms (~70–200 kDa) were also detected in barley leaves infected with powdery mildew (*B. graminis* (DC) Speer f. sp. *hordei*) [31] and supposed to be bacterial or fungal origin [104,105]. The observed increased accumulation of all acidic isoforms of glucanases in infected leaves suggests that these isoforms might act in the extracellular space where they lyse fungal cell walls and limit the spread of disease. It is important to note, however, that the fungus likely contributes to the increase in β-1,3-glucanase content of the pathogen-host system. Up to seven family *GH17* β-1,3-glucanase genes have been identified in *Blumeria graminis* f. sp. *hordei* genome, of which the *bgh03528* was strongly active in conidia [106]. Unfortunately, the literature mostly reports on β-d-glucanhydrolase activities in plants during the first 7 days after fungal infection; data from longer experiments are scarce.

Diversity in β-d-glucanhydrolase activities points to the specificity of enzymes in terms of their ability to release carbohydrates [93,107]. Due to the variable involvement of individual glucanase isoforms in the defence process, it is not possible to establish a clear correlation between glucanase accumulation and established resistance of oat cultivars to *B. graminis* in this study. However, enhanced activity of the 40 kDa isoform in Bay Yan 2 leaves, but not in leaves of other varieties, may be related to the partial non-specific resistance associated with slower pathogen growth as well as the extent of infection, which was confirmed by specific resistance gene evaluation. The detected enzymes might also cleave (at least partly) other β-d-glucans. Our study points to the as yet undescribed accumulation of several hydrolases, which clearly contributes to oat responses to *B. graminis*.

## 5. Conclusions

Uninfected oat varieties formed a relatively homogeneous group according to all measured parameters (β-d-glucan, expression of *CslF* genes, relative activity of β-d-glucan hydrolases, and photosynthetic pigments content). However, after the infection with the leaf fungal pathogen *B. graminis*, changes in all measured parameters were observed. Different reactions of individual oat varieties also appeared, and they were separated into two groups. The varieties Aragon and Vaclav were separated from the others, mainly due to their different expressions of the *CslF6*, *CslF3*, and *CslF9* genes in response to fungal infection. None of the expected relationships between the measured parameters that would confirm the relationship between *CslF* gene expression and β-d-glucan content were statistically significant. Nevertheless, there was at least an indication that they might exist. Different mechanisms probably exist in oats in the synthesis and degradation of β-d-glucan after infection with the used or other fungal phytopathogens that could be further studied.

## Figures and Tables

**Figure 1 polymers-14-03416-f001:**
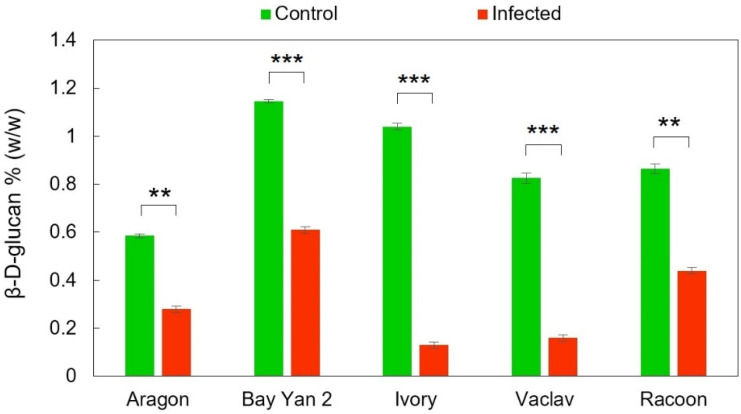
The content of β-d-glucans in oat leaves without and with infection with *B. graminis*. Values represent the average values from three replicates; error bars correspond to the standard deviations; ** represents a statistically significant difference using *t*-test at *p* ≤ 0.01; *** represents a statistically significant difference using *t*-test at *p* ≤ 0.001.

**Figure 2 polymers-14-03416-f002:**
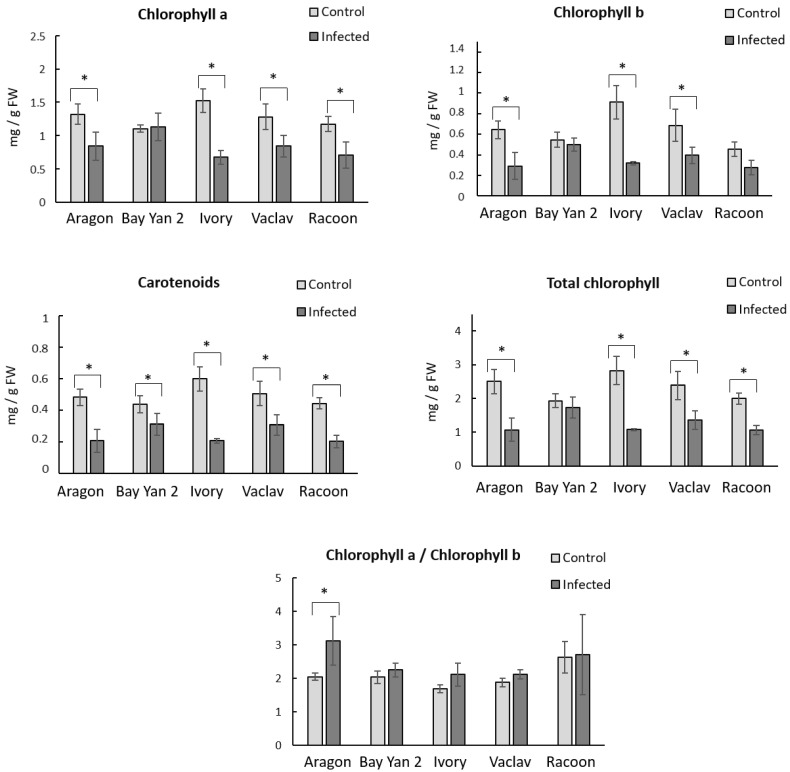
Content of photosynthetic pigments and ratio of Chl_a_ and Chl_b_ in fresh oat leaves (mg/g FW) at control conditions and after infection with *B. graminis*. Values represent the average values from 3 replicates; error bars represent standard deviation; * represents a statistically significant difference between control and infected samples using *t*-test at *p* ≤ 0.05.

**Figure 3 polymers-14-03416-f003:**
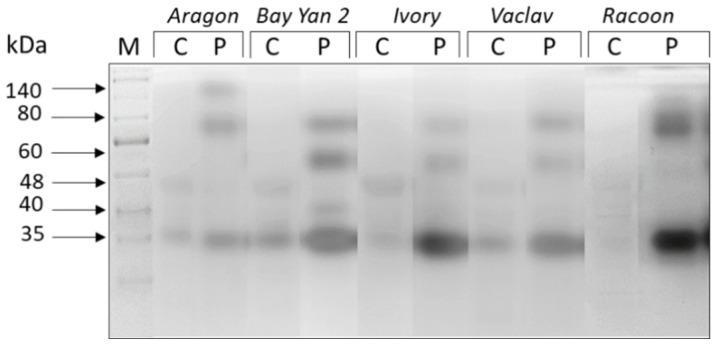
Glucanhydrolases in oat leaves of control (C) and pathogen-infected samples (P). Arrows indicate the estimated molecular mass of detected enzyme isoforms (in kDa); M—marker.

**Figure 4 polymers-14-03416-f004:**
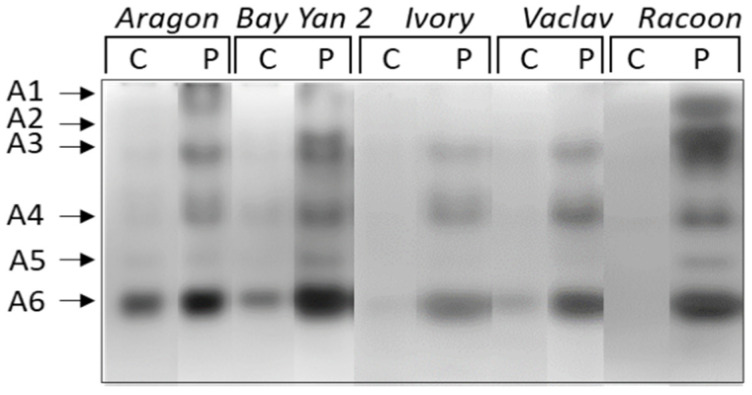
Detection of acidic and neutral isoforms of glucanhydrolases in oat leaves of control (C) and pathogen-infected samples (P). Arrows indicate the β-1,3-glucanase isoforms A1–A6.

**Figure 5 polymers-14-03416-f005:**
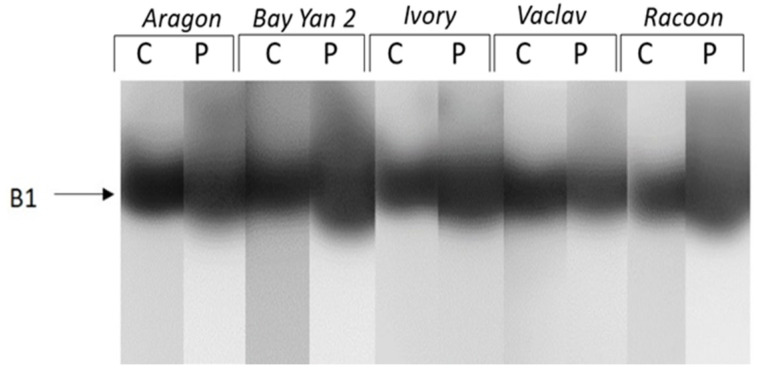
Detection of basic and neutral isoforms of glucanhydrolases in oat leaves of control (C) and pathogen-infected samples (P). Arrow indicates the β-1,3-glucanase isoform B1.

**Figure 6 polymers-14-03416-f006:**
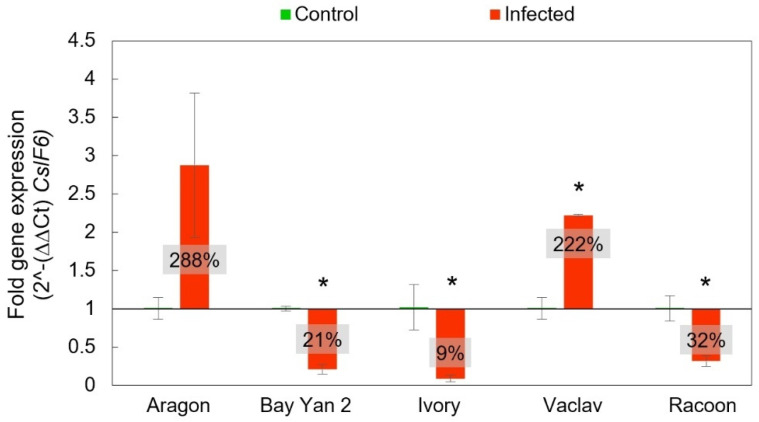
Expression of *CslF6* gene in control and infected leaves based on 2^ -(∆∆Ct). The baseline value 1 represents the control level of expression in control, noninfected plants (i.e., 100%) to which the fold gene expression of the *CslF6* gene from infected oat was related. The values in the box represent fold gene expression in percent—a value above 1 (i.e., above 100%) represents overexpression and values below 1 (i.e., below 100%) represent underexpression of the *CslF6* gene in comparison with the control plants, which were all normalised to the housekeeping gene *EF-1A*. Error bars correspond to the standard deviations; * represents a statistically significant difference between control and infected samples using *t*-test at *p* ≤ 0.05.

**Figure 7 polymers-14-03416-f007:**
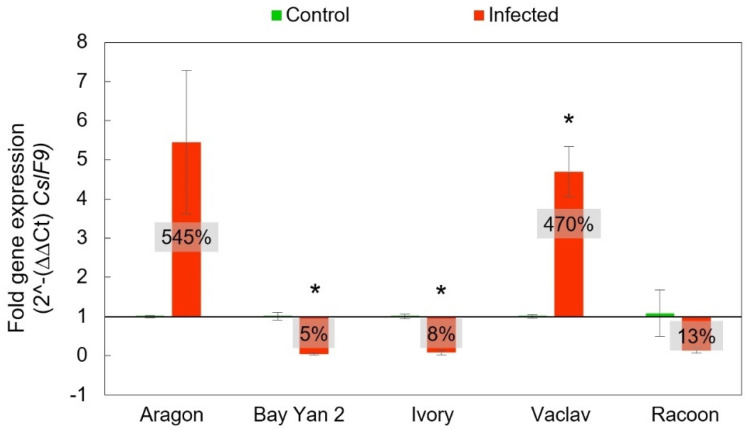
Fold gene expression of *CslF9* gene using RT-qPCR in oat leaves of control and infected samples with *B. graminis* based on 2^-(∆∆Ct) *CslF9*. The baseline from value 1 values in the box represent the same as in Figure 6. Error bars correspond to the standard deviations; * represents a statistically significant difference between control and infected samples using *t*-test at *p* ≤ 0.05.

**Figure 8 polymers-14-03416-f008:**
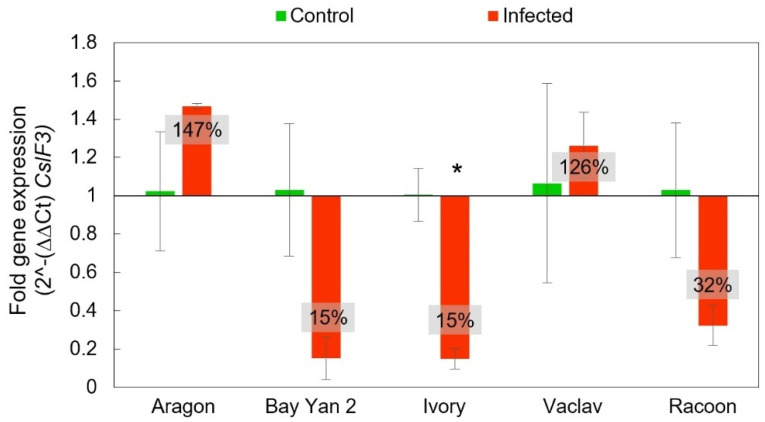
Fold gene expression of *CslF3* gene using RT-qPCR in oat leaves of control and infected samples with *B. graminis* based on 2^-(∆∆Ct) *CslF3*. The baseline from value 1 values in the box represent the same as in Figure 6. Error bars correspond to the standard deviations; * represents a statistically significant difference between control and infected samples using the *t*-test at *p* ≤ 0.05.

**Figure 9 polymers-14-03416-f009:**
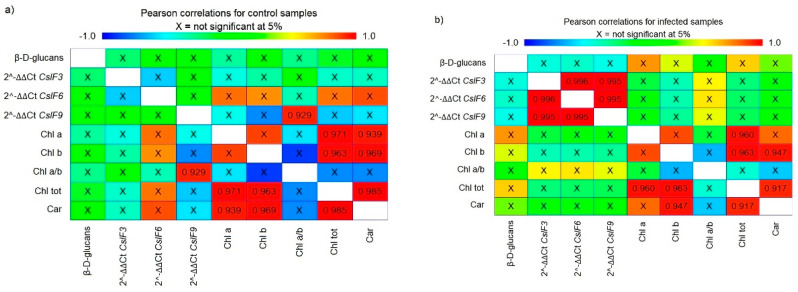
Pearson product–moment correlations between each pair of evaluated parameters in control (**a**) and infected samples with *B. graminis* (**b**). The colour scale changes from dark blue, which represents a correlation value of -1, to red, which represents a value of +1. X represents no statistical difference between the two parameters at 95% significance level. The number shown expresses the value of the correlation between the two parameters and that the correlation is statistically significant at the 95% significance level.

**Figure 10 polymers-14-03416-f010:**
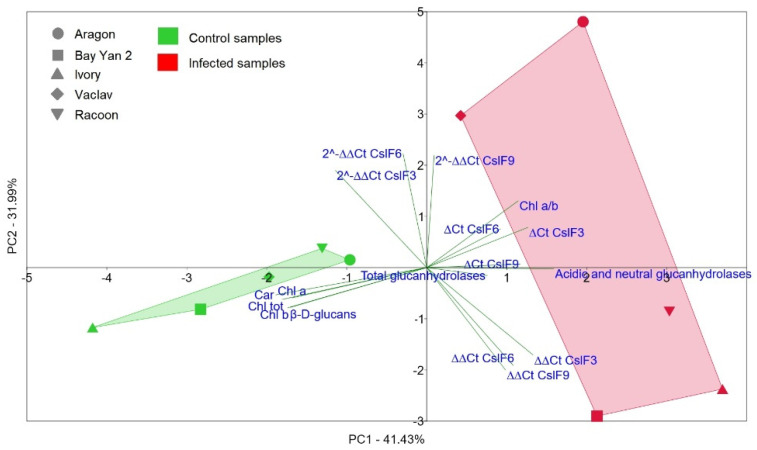
The principal component analysis (PCA) constructed from all measured parameters in oat control and infected samples with *B. graminis*. The PCA graph displays 73.42% of the total variability.

**Table 1 polymers-14-03416-t001:** Primer sequences used for the amplification of partial cDNA products (T_m_—melting temperature).

Gene	Primer Sequence (5′→3′)	PCR Product Length (bp)	T_m_ (°C)
*CslF6*	F: TTGCGCTCGGGATAATGG	91	61
	R: TATACCGATGCTGTGGCAGC		
*CslF3*	F: CAATGTTGATCCGTCGGACC	101	59
	R: TGCCAAGATAAGAGGGCCC		
*CslF9*	F: TTGCGCTCGGGATAATGG	91	61
	R: TATACCGATGCTGTGGCAGC		
*EF-1A*	F: CCAAGAGGCCCTCAGACAAG	101	61
	R: CACTCCGGTCTCAACACGC		

**Table 2 polymers-14-03416-t002:** Relative accumulation of glucanhydrolase isoforms in pathogen-infected oat leaves. The values represent the mean ± standard deviation of three replicates. Bands observed in stressed plants but not in controls are assigned as present; * represents a statistically significant difference using *t*-test at *p* ≤ 0.05; MW—molecular weight; n.d.—not detected.

MW (kDa)	Aragon	Bay Yan 2	Ivory	Vaclav	Racoon
140	present	n.d.	n.d.	n.d.	n.d.
80	present	present	present	present	present
60	present	present	present	present	present
48	0.48 ± 0.08 *	0.36 ± 0.05 *	0.30 ± 0.15 *	0.24 ± 0.05 *	0.17 ± 0.03 *
40	0.87 ± 0.14	3.08 ± 0.59 *	0.87 ± 0.17	1.08 ± 0.22	0.63 ± 0.20 *
35	1.65 ± 0.82	1.09 ± 0.20	4.45 ± 2.53	2.85 ± 1.25	1.68 ± 0.55 *

**Table 3 polymers-14-03416-t003:** Relative accumulation of acidic and neutral isoforms of glucanhydrolases in pathogen-infected oat leaves. The values represent the mean ± standard deviation of three replicates. Bands observed in stressed plants but not in controls are assigned as present; * represents a statistically significant difference using t-test at *p* ≤ 0.05; n.d.—not detected.

Isoform	Aragon	Bay Yan 2	Ivory	Vaclav	Racoon
A1	present	n.d.	n.d.	n.d.	present
A2	n.d.	present	n.d.	n.d.	present
A3	6.16 ± 2.23 *	13.71 ± 6.46 *	3.97 ± 0.72 *	9.16 ± 0.76 *	17.94 ± 11.24 *
A4	2.58 ± 1.18	9.84 ± 4.45 *	2.07 ± 0.67	8.86 ± 3.52 *	7.00 ± 2.37 *
A5	1.44 ± 0.42	8.14 ± 2.89 *	present	present	present
A6	1.71 ± 0.34 *	5.53 ± 2.35 *	13.18 ± 6.22 *	6.25 ± 3.72 *	3.11 ± 1.23

## Data Availability

The data supporting reported results can be provided upon request to the interested individuals/researchers. The data presented in this study are available on request from the corresponding author.

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
