# Peer review of "The Cell-Wall β-d-Glucan in Leaves of Oat (Avena sativa L.) Affected by Fungal Pathogen Blumeria graminis f. sp. avenae"

_polymers, 2022, doi:10.3390/polym14163416_

Round 1

Reviewer 1 Report

The work by Gregusová et al. (The Cell-Wall β-D-Glucan in Leaves Affected by Fungal Pathogen) reports the effect of fungal pathogen B. graminis on the expression of Cs1F6, CslF9, and CslF3 106 genes, the content of β-D-glucan, the relative activity of glucanase, and content of chlorophyll in oat leaf. For this, five oat (Avena sativa L.) varieties (Aragon, Bay Yan 2, Ivory, Vaclav, Racoon) were selected and relevant tests were performed accordingly.

In general, the manuscript is well-organized and easy to follow. I think the outcomes of the report provide valuable information. I recommend the publication of the after some points are given below are addressed.

-          The Abstract section must be rewritten. In the present form, it is really hard to understand the main focus of the report. Also, please provide the main objective of the study and some major results of the study.

-          The last paragraph of the Introduction section must be extended. Please provide more data in the end.

-          I noticed some grammatical errors. Please fix them. 

Author Response

Responses to questions and comments of Reviewer 1

Question/comment 1: The Abstract section must be rewritten. In the present form, it is really hard to understand the main focus of the report. Also, please provide the main objective of the study and some major results of the study.

Response 1: Accepted. Abstract has been completely rewritten. Now the Abstract includes the main objectives of the study and some of the major results obtained.

Question/comment 2: The last paragraph of the Introduction section must be extended. Please provide more data in the end.

Response 2: Accepted. The last paragraph in Introduction (aims) was rewritten.

Question/comment 3: I noticed some grammatical errors. Please fix them. 

Response 3: Accepted. The text was corrected by English native speaker.

Reviewer 2 Report

This manuscript describes changes in b1,3 glucans in oat seedlings’ response to powdery mildew fungus, Blumeria graminis. Five oat strains were tested for responses related to pigment content and b1,3 and mixed b1,3/b1,4glucans.  In general, the results document pathogen-induced decreases in b-glucans, bleaching in most strains, and strain-variable responses in expression of oat glucan synthesis genes. Covariance and PC analyses show correlations between infection and bleaching, but other correlations lack statistical significance.

There are major issues with the manuscript:

1.     It is not clear how many biological replicates of growth and infection were done. The reported growth conditions show different relative humidity for control and infected plants. 

2.     The error bars in determination of glucan content of the leaves (Fig 1) are remarkably small for this kind of analysis. They are stated to be “standard deviations from three replicates.”  In my opinion, these are likely to be technical replicates of the same samples.

3.     The glucanase analysis shows a large increase in activity in infected leaves from all strains. However, there are two essential elements missing:

a.     First, the description of the experimental design is inadequate. Extracts were electrophoresed in the presence of SDS and glucan. The gels were then soaked in acetate buffer, and stained with tetrazolium blue, which detects hydrolysis. The enzymes are reported by MW, apparently from gel mobility.

                                               i.     Neither the text not the original reference describe the effects of adding glucan to the gels. (The original method uses an overlay substrate gel instead of including glucan in the separation gel.)

                                             ii.     The authors did not describe the experimental conditions that make figs 4 and 5 different from fig 3.

                                            iii.     Statements like the following are hard to justify from the presented figures: “Also, higher (8.14-times) accumulation was observed in isoform A5 in infected leaves of Bay Yan 2. This isoform was also detected in infected leaves of Ivory, Vaclav, and Racoon (Figure 4, Table 3).” (lines 285-287). The result is difficult to justify, given the appearance of faint or absent bands in control extracts and the lack of description of background subtraction procedures. This point is exacerbated by the electrophoresis in the presence of glucan, which may increase background staining, because the enzymes hydrolyze substrate as they pass through each region of the gel. 

                                            iv.     No data is presented on the effects of SDS on activity

b.     The authors have not attempted to distinguish between glucanases encoded by the host and those encoded by the pathogen. A bioinformatics search would at least serve to identify how many isozymes might be encoded by host and pathogen genomes. Therefore, it is impossible to know whether the activities are correlated with infection or resistance to infection.

  Issues for data interpretation:

1.     The glucan analysis for infected leaves must include both fungal and plant glucans. Therefore, it is difficult to assess the significance of the pathogen-induced decreases.

2.     The susceptibility of the 5 tested strains to B. graminis is not mentioned. This makes comparison of the responses difficult to assess.

3.     A minor issue is that fold-changes in gene expression and glucanase activities are reported to 4-5 significant figures (“287.73%” in fig 6), when the error bars can only support 1-2 sig figs (~300%).

4.     I find this statement (lines 287 to 291) not to be supported by the accompanying figure “Determination of the accumulation of basic and neutral isoforms was non-reproducible in the chosen detection system and could not be reliably quantified. However, based on partial results, it can be reported at least one basic isoform with a decreasing tendency of accumulation under powdery mildew stress (Figure 5).”

Author Response

Responses to questions and comments of Reviewer 2

Question/comment 1: This manuscript describes changes in b-1,3-glucans in oat seedlings’ response to powdery mildew fungus, Blumeria graminis. Five oat strains were tested for responses related to pigment content and b-1,3 and mixed b-1,3-/b-1,4-glucans. In general, the results document pathogen-induced decreases in b-glucans, bleaching in most strains, and strain-variable responses in expression of oat glucan synthesis genes. Covariance and PC analyses show correlations between infection and bleaching, but other correlations lack statistical significance.

Response 1:

Question/comment 2: It is not clear how many biological replicates of growth and infection were done.

Response 2: Accepted. We got one pot with ten plants of the same variety for control and for infection. We took samples from one pot but not only from one plant but from several from the same batch, therefore we got technical replicates not biological one.

Question/comment 2.1: The reported growth conditions show different relative humidity for control and infected plants. 

Response 2.1: All variants were grown together in the cultivation room until the inoculation date, so they had the same conditions. After inoculation, variants were grown separatelly for 4 weeks because of technical reasons.According the measurements we could made, the relative air humudity was 42 and 54, so there was a difference there. But we are sure, this was not a significant difference influenced the results presented in the manuscript. We hope, the reviewer could accept this explanation.

Question/comment 3: The error bars in determination of glucan content of the leaves (Fig 1) are remarkably small for this kind of analysis. They are stated to be “standard deviations from three replicates.”  In my opinion, these are likely to be technical replicates of the same samples.

Response 3: Yes, they were technical replicates from dry sample. We took fresh leaves from various plants from one pot. We dry them and milled them and from this dry samples from each variety and growth we determined β-D-glucan.

Question/comment 4: The glucanase analysis shows a large increase in activity in infected leaves from all strains. However, there are two essential elements missing:

First, the description of the experimental design is inadequate. Extracts were electrophoresed in the presence of SDS and glucan. The gels were then soaked in acetate buffer, and stained with tetrazolium blue, which detects hydrolysis. The enzymes are reported by MW, apparently from gel mobility.

Neither the text not the original reference describe the effects of adding glucan to the gels. (The original method uses an overlay substrate gel instead of including glucan in the separation gel.)

Response 4: The method used is a modified procedure described by Pan et al., which is based on the presence of the enzyme-substrate directly in the gel. The approach allows the detecting of a more complete enzyme profile with more sharp bands compared to the original methodology, which uses an overlay gel. The advantages of the modified method also include better reproducibility and sensitivity. The method has been successfully used in several previous works (e.g. Hlinková et al., 2002; Fráterová et al., 2013; Maglovski et al., 2017; Michalko et al., 2013; Piršelová et al., 2011; Żur et al., 2013). The same extracts were separated under three conditions (denaturing, native-acidic and native-basic, respectively) to maximize the separation rate and detect differences conditioned by treatments. The corresponding text in the Methods part was edited and we believe in this form it more clearly presents the procedure applied.

Hlinková, E. et al. Chitinases and Endoglucanases Synthesized in the Infected Barley Leaves in the Powdery Mildew Period Sporulation. Plant Protect. Sci., 38, 2002: 469–473

Fráterová, L. et al. The role of chitinases and glucanases in somatic embryogenesis of black pine and hybrid firs. Cent. Eur. J. Biol., 8(12), 2013, 1172-1182. 

Maglovski et al. Nutrition supply affects the activity of pathogenesis-related β-1,3-glucanases and chitinases in wheat. Plant Growth Regulation. 81, 2017, 1-11.

Michalko, J. et al. Glucan-rich diet is digested and taken up by the carnivorous sundew (Drosera rotundifolia L.): Implication for a novel role of plant β-1,3-glucanases. Planta (2013) 238: 715–725.

Piršelová, B. et al. Biochemical and physiological comparison of heavy metal-triggered defense responses in the monocot maize and dicot soybean roots. Mol Biol Rep 38, 3437–3446 (2011).

Żur I. et al. β-1,3-glucanase and chitinase activities in winter triticales during cold hardening and subsequent infection by Microdochium nivale. Biologia 68/2: 241-248, 2013

Question/comment 5: The authors did not describe the experimental conditions that make figs 4 and 5 different from fig 3.

Response 5: The figures present protein profiles of the same biological material (extracts) but separated under different electrophoretic conditions (denaturing, native-acidic, native-basic). The methodology and the figure legends were edited to clarify this issue (please, see the corresponding parts highlighted with RED letters)

Question/comment 6: Statements like the following are hard to justify from the presented figures: “Also, higher (8.14-times) accumulation was observed in isoform A5 in infected leaves of Bay Yan 2. This isoform was also detected in infected leaves of Ivory, Vaclav, and Racoon (Figure 4, Table 3).” (lines 285-287). The result is difficult to justify, given the appearance of faint or absent bands in control extracts and the lack of description of background subtraction procedures. This point is exacerbated by the electrophoresis in the presence of glucan, which may increase background staining, because the enzymes hydrolyze substrate as they pass through each region of the gel.

Response 6: The reviewer is also correct that substrate hydrolysis can occur during separation. This, however, is minimized by running the gel at low temperatures. Still, background subtraction was applied by the ScionImage software; we edited a corresponding more detailed description as follows: „The relative activity of β-1,3-glucanase isoforms was determined based on the background-corrected mean density of fractions (evaluated in pixels) and was expressed as multiple of the control sample.“ 

Concerning the weak and absent bands, we would like to point out their reproducible nature under rather a homogenous background intensity level, which can only be achieved when extracts from all variants are co-separated in the same gel; to stress this issue we edited this information in the Method part.

Further, The method certainly does not quantify the absolute enzyme activities of individual fractions, nevertheless, in sense of the explanations described above, they allow for the calculation of relative change in enzyme activity, attributable to altered experimental conditions

Question/comment 7: No data is presented on the effects of SDS on activity.

Response 7: We assume SDS does not affect enzyme activity due to rigorous washing out from the gel and to renaturation step. If still any inhibitory activity occurs, a certain bias is applicable similarly for individual isoforms in the experimental variants, thus relative change in enzyme activity in response to the treatment is likely unaltered.

Question/comment 8: The authors have not attempted to distinguish between glucanases encoded by the host and those encoded by the pathogen. A bioinformatics search would at least serve to identify how many isozymes might be encoded by host and pathogen genomes. Therefore, it is impossible to know whether the activities are correlated with infection or resistance to infection.

Response 8: We can see the point of the respected reviewer. Indeed, the bioinformatic approach would reveal all putative enzyme isoforms in both host as well as pathogens. However, it would not deliver information on actually accumulated (active) enzymes. For this reason, a protein-based approach is more informative for revealing the (changes of) enzyme profile in organisms. In the gels, enzyme activities can in principle originate from both organisms in the experimental system. In reality, however, the biomass of the pathogen is so low compared to the biomass of the tissue taken for the extraction, that fungal activity can be considered negligible. To confirm this, enzyme activity was detected in protein extracts obtained from 200 mg fungal tissue scraped from leaves, similar to the mass taken for plant protein extraction) (Figure S1). We edited this information to the manuscript (please, see the Method section).

Figure S1. Powdery mildew (left panel) was scraped with a scalpel from leaf material of Racoon variety (200 mg) and proteins were extracted from the spores of fungus (Bg – Blumeria graminis) as well as from a similar amount of leaf tissue (control – C, infected – I) as described. 20 µg proteins were separated in native PAGE with laminarin and detected for the presence of acidic glucanases (right panel). The results from two biological replicates (1, 2) showed that fungal contribution to the measured enzyme values in leaf tissue is negligible.

Issues for data interpretation:

Question/comment 9: The glucan analysis for infected leaves must include both fungal and plant glucans. Therefore, it is difficult to assess the significance of the pathogen-induced decreases.

Response 9: We used for determination megazyme assay kit for cereal 1,3-1,4-β-D-glucan. The same company also has kit for fungal 1,3-1,6-β-D-glucan and chemicals and protocol is different from  cereal  one,  therefore we eliminate this problem.

Question/comment 10: The susceptibility of the 5 tested strains to B. graminis is not mentioned. This makes comparison of the responses difficult to assess.

Response 10: We tested resistance genes Pm 1, Pm 2 and Pm 3 but none of 5 varieties has this gene for resistance against B.  graminis. Plants after four weeks were fully covered by this fungal pathogen, and it was visible that each of varieties were susceptible to B. graminis.

Question/comment 11: A minor issue is that fold-changes in gene expression and glucanase activities are reported to 4-5 significant figures (“287.73%” in fig 6), when the error bars can only support 1-2 sig figs (~300%).

Response 11: We have once again gone through all the calculations and the corresponding statistics, and it is correctly stated in the manuscript. The impact on the result of the statistics is primarily on the variability of the measured data in individual replicates. In our case in Figure 6 and the variety Aragon, a large variability was found in the infected replicates, which did not show any statistical difference when compared to the control. We were surprised by this result and therefore we made additional statistical evaluations to verify this result. All additional evaluations, such as the F-test and the Mann-Whitney (Wilcoxon) W-test were also insignificant, and we have also reported this in chapter 3.4. The result was similar for the CslF9 gene in Aragon variety (Figure 7), with a statistically significant difference only in the F-test. We also present this information in the text in chapter 3.5.

The other varieties, although with a smaller difference in the fold gene expression compared to the controls, showed a statistical difference and it is again due to the variability of the measured data in individual replicates. In the case of these varieties, which showed significant differences, the variability of the data during repeated measurements was much smaller, and this small variability had effect on the result of the statistical comparison.

Question/comment 12:  I find this statement (lines 287 to 291) not to be supported by the accompanying figure “Determination of the accumulation of basic and neutral isoforms was non-reproducible in the chosen detection system and could not be reliably quantified. However, based on partial results, it can be reported at least one basic isoform with a decreasing tendency of accumulation under powdery mildew stress (Figure 5).”

Response 12: We agree with the respected reviewer. The sentences were modified to strictly express our observations as follows: „A single basic/neutral isoform was reproducibly present in experimental plants; its responsiveness to powdery mildew, however, was not confirmed statistically (Figure 5).“

Reviewer 3 Report

Dear authors

The manuscript has the potential to be accepted after correcting and answering these points:

General comment: Extensive English language corrections, including spelling, grammar, and punctuation, are recommended.

Title:

The scientific names of plants and fungi should be included in the title.

Abstract:

The goal should be stated explicitly, as well as the precise goal of this investigation. In other words, what are the benefits of this research? The order of the findings in the abstract should be the same as it is in the Results section.

L25-27: The proportion of increasing and decreasing gene expression should be included.

Keywords:

Please omit the terms used in the title formation.

Introduction:

General observation: this part is poorly written. There are various brief sentences that are written separately and must be joined. 

L76-79: the gene Hv……. regulation: this sentence should be deleted#

L81-83: During……invasion: This sentence is meaningless and should be eliminated.

L91: resistant cultivar of what? The name of plant should be inserted

Materials and methods

L118: What made you select the third leaf stage?

The procedure of fungal infecting oats is not explained clearly. This section need detailing and rewriting to be improved.

L123-124: Why was RNA stored in ethanol? Please double-check it.

L125: β-D-glucan determination in leaves: This section should be comprehensive.

L145-150: This part is poorly written; the sentences should be properly structured and integrated.

The authors should describe the SDS-PAGE preparation procedure in detail.

The RNA extraction technique should be specified or a reference should be provided.

Results:

L225-226: How did you compute the decreasing or reduction percentage? Mention it in statistical data analysis. I believe there is an error in the calculation.

L231: The authors should include the calculated and critical (p-value) t-test values in the figure 1 caption.

Table 2 and Figure 3: The results of table 2 and Figure 3 do not correspond. Figure 3 shows that there is no detectable band for isoform 48 kD for the Aragon variety on the SDS-OAGE gel. The same problems exist for other isoforms. Because the SDS-PAGE gel lacks net and clear bands, this result is inaccurate. It is preferable to replace it with another clear gel or to remove it from your results.

I have a question about the gel SDS-PAGE profile. You extracted the total protein from the oat leaves; how did you know these bands represented Glucanhydrolase isoforms?

Furthermore, you had to use a standard protein in a variety of concentrations, such as BSA, and place it on the gel alongside your samples. Using the IMAGE J software and the intensity of bands of standard protein, you can quantify and determine the increasing or decreasing percentage of Glucanhydrolase accumulation. There is also an issue with the protein ladder; as you mentioned in L162, the range of the protein ladder is 10 to 260, but I do not see this range on the gel.

L303-305: This phrase should be included in the statistical data analysis.

L305: Describe the problem here. The gene expression was increased approximately threefold when compared to the control, explaining why the difference between infected and uninfected plants is not significant. To better understand gene expression levels, I believe it is preferable to use raw RT-PCR data rather than percentage values. I have a question about this point: how did you conduct the statistical analysis using the percentage values? The results in Figures 6, 7, and 8 are therefore incorrect and should be recalculated using the original data. You can mention the percentage of increasing and decreasing gene expressions in the context of the manuscript after recalculation.

L307: change P ≥ 0.05 to P ≤ 0.05

L380: The interpretation of the PCA plot is unclear and should be rewritten. It is preferable to create two separate PCA plots: one for the control group and one for the infected group.

Discussion:

It is insufficient. It's just another review. The authors should interpret the most significant findings here. For example

what are the reasons for the lack of statistical significance between β-D-glucans 28 content and CslF gene expression?

Considerable over-expression of CslF6, CslF9, and CslF3 genes was recorded in two 442 oat varieties (Vaclav and Aragon) after the infection. However, the other three varieties 443 (Bay Yan 2, Ivory, and Racoon) showed reduced expression of these genes

Best regards

Author Response

Responses to questions and comments of Reviewer 3

Question/comment 1:  General comment: Extensive English language corrections, including spelling, grammar, and punctuation, are recommended.

Response 1: Accepted. The text was corrected by English native speaker.

Question/comment 2:  Title: The scientific names of plants and fungi should be included in the title.

Response 2: Accepted.

Question/comment 3:  Abstract: The goal should be stated explicitly, as well as the precise goal of this investigation. In other words, what are the benefits of this research? The order of the findings in the abstract should be the same as it is in the Results section.

Response 3: Accepted. The last paragraph in Introduction (aims) was rewritten.

Question/comment 4:  L25-27: The proportion of increasing and decreasing gene expression should be included.

Response 4: Accepted. However, the Abstract is limited to only 200 words.

Question/comment 5:  Keywords: Please omit the terms used in the title formation.

Response 5: Accepted.

Introduction:

Question/comment 6: General observation: this part is poorly written. There are various brief sentences that are written separately and must be joined. 

Response 6: In the Introduction part, mostly long sentences are used. We rather expected a comment that long sentences should be shortened. This is usually requested by Reviewers.

Question/comment 7: L76-79: the gene Hv……. regulation: this sentence should be deleted

Response 7: Accepted. The sentence is deleted

Question/comment 8: L81-83: During……invasion: This sentence is meaningless and should be eliminated.

Response 8: Accepted. The sentence is deleted

Question/comment 9: L91: resistant cultivar of what? The name of plant should be inserted

Response 9: accepted. Plant (wheat, Triticum aestivum L.) is added to this sentence.

Materials and methods

Question/comment 10: L118: What made you select the third leaf stage?

Response 10: Thank you for your question. We decided the 3rd leaf stage is BBCH13 according to the Zadoks scale, this is when the seedling stage begins/may begin, i.e., the end of the seedling stage, it is the transition from the juvenile stage to the adult stage. We also need to consider technical issue with larger plants, when infection could be more difficult to get under inoculation tower and inoculation would not be as uniform as it should be.

Question/comment 11: The procedure of fungal infecting oats is not explained clearly. This section need detailing and rewriting to be improved.

Response 11: Accepted. Thank  you  for you  note. Sentence was added. Plants were cultivated to the third leaf stage and in this stage, they were inoculated with spores (1000-2500 spores per cm2) of B. graminis by using settling tower. Spores were dispersed in turbulent air in settling tower according methods Schwarzbach et al. (2006). Density of inoculation was determined on Petri dishes with agar medium which were placed on the top of inoculated pots.

Question/comment 12: L123-124: Why was RNA stored in ethanol? Please double-check it.

Response 12: Accepted. Dear reviewer, you were right, it was wrong formulation of sentence, Leaves samples for RNA isolation were stored in ethanol. You can use RNAlater stabilization solution, but ethanol also works good for stabilize RNA in plant sample and its cheaper.

Question/comment 13: L125: β-D-glucan determination in leaves: This section should be comprehensive.

Response 13: Accepted. The protocol for determination of β-D-glucan is rewritten focused on details of process.

Question/comment 14: L145-150: This part is poorly written; the sentences should be properly structured and integrated.

Response 14: Accepted.

Question/comment 15: The authors should describe the SDS-PAGE preparation procedure in detail.

Response 15: SDS-PAGE gel preparation is a standard procedure described in detail in Maniatis et al (1982). To keep the text on the experimental procedure concise, we take the freedom to edit this reference instead of a detailed description. In case the respected reviewer insists, however, we are ready to edit it.

Maniatis, T., Fritsch E.F. and Sambrook, J.K. (1982) Molecular Cloning: A Laboratory Manual. Cold Spring Harbor Laboratory, Cold Spring Harbor.

Question/comment 16: The RNA extraction technique should be specified or a reference should be provided.

Response 16: Thank  you  for your note. I add reference with protocol of extraction  of RNA by Trizol method.

Chomczynski P, Sacchi N: Single-step method of RNA isolation by acid guanidinium thiocyanate-phenolchloroform extraction. Anal Biochem 162: 156–159, 1987

Results:

Question/comment 17: L225-226: How did you compute the decreasing or reduction percentage? Mention it in statistical data analysis. I believe there is an error in the calculation.

Response 16: The fold gene expression we calculated with comparative CT method based on this reference: Schmittgen, T. D.; Livak, K. J. Analyzing real-time PCR data by the comparative CT method. Nat Protoc 2008, 3, 1101–1108. https://doi.org/10.1038/nprot.2008.73. This is stated at the end of chapter 2.5 and we think that listing the reference is sufficient and it is not necessary to detail the entire calculation procedure. However, we recalculated everything and found no error in the calculation, and it is stated correctly in the manuscript.

Question/comment 18: L231: The authors should include the calculated and critical (p-value) t-test values in the figure 1 caption.

Response 17: The P-values for oat varieties we have added directly into the text in chapter 3.1 and in the Figure 1 caption we left it in the original wording marked as "*" and "**" for clarity.

Question/comment 19: Table 2 and Figure 3: The results of table 2 and Figure 3 do not correspond. Figure 3 shows that there is no detectable band for isoform 48 kD for the Aragon variety on the SDS-OAGE gel. The same problems exist for other isoforms. Because the SDS-PAGE gel lacks net and clear bands, this result is inaccurate. It is preferable to replace it with another clear gel or to remove it from your results.

Response 19: We can see the point of the respected reviewer; the gel images in the PDF might show the given bands as invisible. Nevertheless, the values on the images were sensed and quantified by software with a sensitivity exceeding that of the naked eye. At the same time we believe, the bands can be more visible when (in a positive case) the figures will be processed for printing compared to the small-sized figure in the generated PDF.

Question/comment 20: I have a question about the gel SDS-PAGE profile. You extracted the total protein from the oat leaves; how did you know these bands represented Glucanhydrolase isoforms?

Response 20: The bands detected we assume as glucanhydrolases given that, among the many protein fractions separated in gels, only some (at the given positions) were able to hydrolyze the laminarin in the gel; laminarin is a specific substrate of β-1,3-glucanases, commonly used in similar enzyme assays.

Question/comment 21: Furthermore, you had to use a standard protein in a variety of concentrations, such as BSA, and place it on the gel alongside your samples. Using the IMAGE J software and the intensity of bands of standard protein, you can quantify and determine the increasing or decreasing percentage of Glucanhydrolase accumulation. There is also an issue with the protein ladder; as you mentioned in L162, the range of the protein ladder is 10 to 260, but I do not see this range on the gel.

Response 21: We can see the point of the respected reviewer, unfortunately, quantification of glucanase activities using a standard protein band is not possible because BSA has no enzyme activity and can not be detected after staining with TTC. Furthermore, in the profiles of enzyme activities detected with this method, there are often isoforms that show highly similar activities throughout all the variants (please see e.g. Maglovski et al., 2017), supporting the standardized setting of this approach. The method yields data which have previously been shown to correlate well with gene expression or other physiological data.

Maglovski et al. Nutrition supply affects the activity of pathogenesis-related β-1,3-glucanases and chitinases in wheat. Plant Growth Regulation. 81, 2017, 1-11.

Nevertheless, loading the same amount of tissue protein extracts, as seen, results in data with reasonable dispersion from the same variants, and allows for relative quantification of detected activities. Though we agree with the reviewer that our data can not be considered absolute enzyme activity values, their relative change can be determined reliably.

Concerning the protein ladder used we would like to point out that the values indicated refer to the estimated size of enzyme isoforms detected, not of the components of the marker mixture. We stressed this in the legends accordingly. At the same time, the amount of the marker loaded resulted in rather poor visibility of some of the fractions, while the smallest molecules (≤ 10 kDa) likely have run out of the gel already; this is the reason why the indicated range intuitively does not fit with the observed ladder.

Question/comment 22: L303-305: This phrase should be included in the statistical data analysis.

Response 22: Dear reviewer, thank you fr this comment and suggestion. You are right, the sentence could be in the part of statistical data analyses. But we would like to keep the sentence as it is, because we think, it could also stay as i tis now. We hope, you could accept our conditon. In this state we think, the sentence is suitable.  

Question/comment 23: L305: Describe the problem here. The gene expression was increased approximately threefold when compared to the control, explaining why the difference between infected and uninfected plants is not significant. To better understand gene expression levels, I believe it is preferable to use raw RT-PCR data rather than percentage values. I have a question about this point: how did you conduct the statistical analysis using the percentage values? The results in Figures 6, 7, and 8 are therefore incorrect and should be recalculated using the original data. You can mention the percentage of increasing and decreasing gene expressions in the context of the manuscript after recalculation.

Response 23: The calculation of fold gene expression was based on the original raw data from RT-PCR, and from the beginning it applied to the housekeeping gene, both control and infected samples. In the first step, we calculated average Ct value for technical replicates of each sample from raw Ct values. The next step we calculated delta Ct for each control and infected sample using formula: ∆Ct = Ct (CslF6, CslF9 or CslF3) - Ct (EF-1A). After calculating the ∆Ct Control average, we calculated the ∆∆Ct values for each sample using formula: ∆∆Ct = ∆Ct (our sample) – ∆Ct (Control average). Finally, we calculated the fold gene expression values using 2 to the power of negative ∆∆Ct as follows: Fold gene expression = 2^-(∆∆Ct). For clarity, we expressed these results in percentages in graphs, but we statistically evaluated the data that corresponded to 2^-∆∆Ct which were calculated from raw RT-PCR data, not percentages. However, we recalculated everything and found no error in the calculation and statistics, and it is stated correctly in the manuscript. When calculating, we followed Livak & Schmittgen 2001 (doi:10.1006/meth.2001.1262) as well as Schmittgen & Livak 2008 (doi:10.1038/nprot.2008.73). We stated the information about the percentages in Figure 6 caption and it means that fold change of 1 means 100% in control condition and a fold-change value above 1 in our gene of interest showed upregulation relative to the control (2.87- fold change = 287% gene expression relative to control, 2.21 = 221%, etc.). Values below 1 are indicative of gene downregulation relative to the control (fold change of 0.31 is 31% gene expression relative to control, etc.).

Question/comment 24: L307: change P ≥ 0.05 to P ≤ 0.05

Response 24: In the manuscript, it is stated correctly because in the given sentence we write that there was no statistical difference detected and therefore there is P ≥ 0.05. Also as i tis written in sentences in the next chapter of the Results part. 

Question/comment 25: L380: The interpretation of the PCA plot is unclear and should be rewritten. It is preferable to create two separate PCA plots: one for the control group and one for the infected group.

Response 25: Thanks for the reminder, but let's gently disagree and we keep only 1 PCA graph with control and infected samples. Because only in this way, in one graph, we can see how the controls are divided from the infected samples. At the same time, we think that the current statement regarding the result of the PCA analysis is sufficient, and our statements are also supported by Figure 10.

Discussion:

Question/comment 26: It is insufficient. It's just another review. The authors should interpret the most significant findings here.

For example: what are the reasons for the lack of statistical significance between β-D-glucans 28 content and CslF gene expression?

Considerable over-expression of CslF6CslF9, and CslF3 genes was recorded in two 442 oat varieties (Vaclav and Aragon) after the infection. However, the other three varieties 443 (Bay Yan 2, Ivory, and Racoon) showed reduced expression of these genes

Response 27: Although we did not make fundamental changes to the text of the Discussion, we did make some additions/corrections (also to your comments).

Round 2

Reviewer 2 Report

This revised manuscript addresses most of the reviewers’ points. However, some major issues have not been resolved (see comments 8, 9 and 11 below). There are also minor revisions requested in points 4.2 and 10.  Listed in order and number of the “Question/Comment numbers in the Authors’ Response:

2.     The results are based on a single growth experiment, and thus reflect a limited set of biological variations. The Methods and procedures now adequately describe this limitation.

3.     The experiment is now accurately described.

4.1 The experimental design is now adequately described.

4.2 The references in the Authors’ Response should be added to the manuscript.

5.   The experiment is now adequately described.

6.   The response is appropriate.

7.   The response is appropriate.

8.   This objection has NOT been adequately answered.  

A. The authors have not used a bioinformatic approach to suggest fungal or host identities for the glucanases. The CAZY database should facilitate this analysis.

B. Also, the authors have presented an argument that the fungus is a small part of the biomass and is therefore not the source of glucanases. This argument is spurious because expression levels can be high when the biomass is low.

C. Supplementary Fig. 1 merely highlights the problem. The fungal tissue shows the same bands as the infected tissue in Fig. 4, whereas these enzymes are absent from the uninfected leaf tissue. It is more likely that the fungus injects glucanases into the plant than that the plant injects them into the pathogen. This result and these interpretations imply that the glucanases are of fungal origin. Together, these observations make the added language on lines 315-316 an erroneous and misleading statement. 

I therefore suggest that the language in the article be changed to reflect an increase in glucanase content of the pathogen-host system, rather than a change in plant response.

9.   Again the authors’ response is not assessable in the absence of data. Both kits would produce positive results on glucan from the other organism, because b1,3 glucanase is a major component of each kit. The authors should change the wording to reflect the uncertainty of the results.

10. This data should be included in the manuscript.

11.  The recalculation of fold-differences in the figures and discussion of glucanase activity and PCR results has NOT addressed the issue. For the RT-PCR measurements and the glucanase activity assays, the results are reported to much greater precision than the variance justifies. Therefore, these numbers (e.g. in Fig. 6, the fold increase for the Aragon strain is listed as “287.73% ± ~180%”). The error bars would only justify stating that this increase is ~290% ± 180%. Given the magnitude of the variance, 5-place precision in reporting is not justified. The authors should be sure not to imply false precision in the results.

12.  The response is adequate.

Finally, the manuscript should be edited for English usage.

Author Response

This revised manuscript addresses most of the reviewers’ points. However, some major issues have not been resolved (see comments 8, 9 and 11 below). There are also minor revisions requested in points 4.2 and 10.  Listed in order and number of the “Question/Comment numbers in the Authors’ Response:

Dear reviewer, we would like to express you a big thank you for your time and energy to carefully read the manuscript and thanks to your comments to increase the quality of the article. I am sorry for any inconveniences during the reviewing process.

Question/Comment 2: The results are based on a single growth experiment, and thus reflect a limited set of biological variations. The Methods and procedures now adequately describe this limitation.

Authors´Response 2: Thank you very much.

Question/Comment 3:     The experiment is now accurately described.

Authors´Response 3: Thank you very much.

Question/Comment 4.1: The experimental design is now adequately described.

Authors´Response 4.1: Thank you very much.

Question/Comment 4.2: The references in the Authors’ Response should be added to the manuscript.

Authors´Response 4.2: Dear reviewer, thank you for your note, references were added to the. Manuscript.  

L219, .. which was successfully used in several previous works [31,46–50].

Question/Comment 5:   The experiment is now adequately described.

Authors´Response 5: Thank you very much.

Question/Comment 6:   The response is appropriate.

Authors´Response 6: Thank you very much.

Question/Comment 7:   The response is appropriate.

Authors´Response 7: Thank you very much.

Question/Comment 8:   This objection has NOT been adequately answered.

  1. The authors have not used a bioinformatic approach to suggest fungal or host identities for the glucanases. The CAZY database should facilitate this analysis.
  2. Also, the authors have presented an argument that the fungus is a small part of the biomass and is therefore not the source of glucanases. This argument is spurious because expression levels can be high when the biomass is low.
  3. Supplementary Fig. 1 merely highlights the problem. The fungal tissue shows the same bands as the infected tissue in Fig. 4, whereas these enzymes are absent from the uninfected leaf tissue. It is more likely that the fungus injects glucanases into the plant than that the plant injects them into the pathogen. This result and these interpretations imply that the glucanases are of fungal origin. Together, these observations make the added language on lines 315-316 an erroneous and misleading statement.

Authors´Response 8: We thank the reviewer for the constructive comments. We admit, our system does not differentiate the origin of enzyme activity in the infected tissue. As our argumentation using Fig.S1 is not acceptable, we skip it from the manuscript. To address the concerns raised, we adjusted the language of indicated section to eliminate the erroneous and misleading statements (please, see the edits made with red color). Though we did not perform bioinformatic approach, we refer the reader to the study in which genome survey and corresponding β-1,3-glucanase analyses were done on Blumeria graminis f. sp. hordei conidia. We believe the respected reviewer finds our statements correct. 

Issues for data interpretation:

I therefore suggest that the language in the article be changed to reflect an increase in glucanase content of the pathogen-host system, rather than a change in plant response.

Question/Comment 9:   Again, the authors’ response is not assessable in the absence of data. Both kits would produce positive results on glucan from the other organism, because b1,3 glucanase is a major component of each kit. The authors should change the wording to reflect the uncertainty of the results.

Authors´Response 9: Dear reviewer, thank you very much for this comment. According to the information on the web page of megazyme (https://support.megazyme.com/support/solutions/folders/8000077001/page/1?url_locale=) it is written: “This kit is a highly specific enzymatic procedure for the measurement of cereal 1,3:1,4-β-D-glucans e.g. from oat, barley and wheat.” And “The cereal β-glucans (so-called mixed-linkage β-glucans) are linear polysaccharides in which D-glucosyl residues are linked 1,3-β- and 1,4-β-, and the ratio of these linkage types varies with the source of the β-glucan (e.g., oats, barley, and wheat).“ Therefore we are sure that the kit and the measurement we used is sufficient for the determination of cereal beta-D-glucan. Also what is written on the megazyme web page: “The structures of mushroom and fungal β-glucans are different from the cereal β-glucans. Fungal and yeast cell walls are composed of ≤ 50% 1,3:1,6-β-glucan, whereas cereal glucans are 1,3:1,4-β-D-glucans. Two enzymes are employed for the hydrolysis of mixed linkage β-glucans in K-BGLU. Lichenase is highly specific and hydrolyses only the β-1,4-linkage next to a β-1,3-. β-glucosidase has varying specificity for β-1,4 and β-1,3 glycosidic bonds. Therefore β-glucosidase will act on the (1,3) chains of 1,3:1,6-β-glucan until it reaches branching. We can estimate that 5-10% of 1,3:1,6 can be measured depending on the structure of β-glucan. 1,3:1,6 β-glucans do not occur naturally in cereal samples.”

According all these information we are sure that our results are correct and in the case that we captured with our cereal glucan kit any of mushroom or yeast glucan (1-3- linkages) in oat samples, all these samples were analysed with the same error, in the same proportion 5-10%. On the other hand, some published papers (for example Martin et al., https://link.springer.com/article/10.1007/s10658-018-1506-8, Havrlentová et al. https://www.mdpi.com/2076-2607/9/10/2108) used the same experimental methods and were successfully published.

We hope, that you, dear reviewer, can accept our explanation and also can accept our results presented in the text.

Question/Comment 10: This data should be included in the manuscript.

Authors´Response 10: Dear reviewer, thank you for your note. This data is included in text.

L141 None of these varieties contain any of the resistance genes Pm1, Pm2, Pm3, and Pm6 against B. graminis determined using the leaf segments inoculated by pathogen isolates [41].

Question/Comment 11.  The recalculation of fold-differences in the figures and discussion of glucanase activity and PCR results has NOT addressed the issue. For the RT-PCR measurements and the glucanase activity assays, the results are reported to much greater precision than the variance justifies. Therefore, these numbers (e.g. in Fig. 6, the fold increase for the Aragon strain is listed as “287.73% ± ~180%”). The error bars would only justify stating that this increase is ~290% ± 180%. Given the magnitude of the variance, 5-place precision in reporting is not justified. The authors should be sure not to imply false precision in the results.

Authors´Response 11: We have rounded the values regarding the fold gene expression and glucanase activity to whole numbers in the entire manuscript. At the same time, for the fold gene expression, we also added the SD values in percentages to the rounded values so that it is clearer to know the variance. Figures 6, 7, and 8 have been edited to round values of % compared to original version.

Question/Comment 12.  The response is adequate.

Authors´Response 12: Thank you very much.

Finally, the manuscript should be edited for English usage.

Thank you for this comment. According to our possibilities, the English was edited.

Reviewer 3 Report

Dear Authors

I appreciate your reply. There are still errors in your manuscript:

L287-L288 (*represents statistically significant difference using t-test, P≤0.01; **represent statistically significant difference using t-test, P≤0.001): This is an incorrect sentence. The correct answer is:

* Represents a statistically significant difference using t-test at P ≤ 0.05

** Represents a statistically significant difference using t-test at P ≤ 0.01

**** Represents a statistically significant difference using t-test at P ≤ 0.001

Please correct errors throughout the entire document.

I disagree on the following points:

Response 17 Because for each asterisk has a value, we can rely on this value and use * or **.

Response 24: If your sentence structure is as follows: there was no statistical difference, the p-value should be written as p > 0.05 rather than p ≥ 0.05, because a p-value of 0.05 (p-value = 0.05) indicates a significant difference

Best regards

Author Response

Dear rewiever, thank you very much for your time and energy to increase the quality of the manuscript. I am sorry for the mistakes made in the manuscript. 

Comment 1: I appreciate your reply. There are still errors in your manuscript:

L287-L288 (*represents statistically significant difference using t-test, P≤0.01; **represent statistically significant difference using t-test, P≤0.001): This is an incorrect sentence. The correct answer is:

* Represents a statistically significant difference using t-test at P ≤ 0.05

** Represents a statistically significant difference using t-test at P ≤ 0.01

**** Represents a statistically significant difference using t-test at P ≤ 0.001

Please correct errors throughout the entire document.

Authors response 1: We have corrected these sentences in the entire document and in all Figure captions which there were relevant. Also, we have edited Figure 1 and corrected the number of asterisks in it.

Comment 2: I disagree on the following points:

Response 17 Because for each asterisk has a value, we can rely on this value and use * or **.

Authors response 2: In the entire document, we have unified the number of asterisks according to the corresponding significance level, i.e. according to your previous comment.

Comment 3: Response 24: If your sentence structure is as follows: there was no statistical difference, the p-value should be written as p > 0.05 rather than p ≥ 0.05, because a p-value of 0.05 (p-value = 0.05) indicates a significant difference.

Authors response 3: We have corrected this P-value for insignificant differences in the entire document.

Round 3

Reviewer 2 Report

The new version of the manuscript has answered my queries. It should now be published. One minor correction: "graminis" should be lower case in the title